# Heart Rate Variability and Sympathetic Activity Is Modulated by Very Low-Calorie Ketogenic Diet

**DOI:** 10.3390/ijerph19042253

**Published:** 2022-02-16

**Authors:** Rita Polito, Anna Valenzano, Vincenzo Monda, Giuseppe Cibelli, Marcellino Monda, Giovanni Messina, Ines Villano, Antonietta Messina

**Affiliations:** 1Department of Clinical and Experimental Medicine, University of Foggia, 71100 Foggia, Italy; rita.polito@unicampania.it (R.P.); anna.valenzano@unifg.it (A.V.); giuseppe.cibelli@unifg.it (G.C.); giovanni.messina@unifg.it (G.M.); 2Department of Experimental Medicine, Section of Human Physiology and Unit of Dietetics and Sports Medicine, University of Campania “Luigi Vanvitelli”, 80138 Naples, Italy; vincenzo.monda@unicampania.it (V.M.); marcellino.monda@unicampania.it (M.M.); ines.villano@unicampania.it (I.V.)

**Keywords:** obesity, visceral adipose tissue, central nervous system, heart rate variability (HRV), weight loss, very low-calorie ketogenic diet (VLCKD), low calorie diet (LCD)

## Abstract

Obesity is characterized by an energy imbalance and by the accumulation of visceral adipose tissue. The energy balance is controlled by a complex set of balanced physiological systems that provide hunger and satiety signals to the brain and regulate the body’s ability to consume energy. The central nervous system controls the metabolic state, influencing the activity of other systems and receiving information from them. Heart rate variability (HRV) is the natural variability of the heart rate in response to several factors. HRV is related to the interaction between the SNS and the parasympathetic. In the light of this evidence, the aim of this study is to investigate the possible effects of the two different dietary regimens such as very low-calorie ketogenic diet (VLCKD) vs. low caloric diet (LCD), on the functions of the nervous system, with particular attention to the autonomous control of heart rate variability (HRV). A total of 26 obese subjects underwent diet therapy in order to reduce body weight; they were also randomly divided into two groups: the VLCKD group and the LCD group. Our results showed that in both groups, there is a reduction in heart rate as an indicator of sympathetic activity; we found a statistically significant variation only in the VLCKD group. Therefore, this study supports the notion that the sympathovagal balance can be modulated by a specific diet, but further studies are needed to clarify the molecular pathway undergoing this modulation.

## 1. Introduction

Obesity is a constantly growing disease in developed countries and is an important risk factor for several cardiovascular and metabolic disorders such as hypertension, type 2 diabetes, dyslipidemia, atherosclerosis, and even certain types of cancer [1]. The World Health Organization has highlighted how obesity is an important social problem, estimating that there is a total of 300 million obese people (BMI > 30, Body Mass Index expressed in kg/m^2^) and more than 1 billion overweight people (BMI > 24.9). For this reason, research into the understanding of the mechanisms underlying body weight control is becoming increasingly important and aims to implement new strategies to reduce the incidence of overweight and obesity [2]. Genetic predisposition, together with inactive lifestyles and a high calorie intake, leads to an imbalance in the management of energy metabolism by the body. The amount of excess energy that is not used can be stored (typically in the form of fat), and if the storage is excessive, you will encounter obesity. The energy balance is controlled by a complex set of balanced physiological systems that provide hunger and satiety signals to the brain and regulate the body’s ability to consume energy [3]. Many anatomical districts can perform different functions in this context and the central nervous system (CNS) certainly plays a dominant role in controlling the metabolic state, influencing the activity of other systems and receiving information from them. The hypothalamus is subjected to a continuous flow of signals from the upper brain areas and the periphery [4]. Adipose tissue plays an important role in the overall regulation of energy metabolism; it is anatomically and functionally distinguished into white adipose tissue and brown adipose tissue (BAT) [5]. 

For many years, nutritional intervention studies have focused on reducing food fat with few positive long-term results. It is now accepted internationally that carbohydrates form the basis of the food pyramid in a healthy diet; however, recent studies have shown that low-carbohydrate diets could be a much faster and more effective way to lose weight. Therefore, one of the most researched weight loss strategies in recent years is the ketogenic diet. It is a nutritional regime based on the greatly reduced intake of carbohydrates, resulting in what is called “dietary ketosis”, a physiological condition of the increased production of ketone bodies. Ketone bodies, produced at liver level through the activation of a metabolic pathway such as that used by the oxidation of fatty acids, are conveyed to extra-hepatic tissues (such as the brain) to be used as an alternative energy source; this allows the metabolism to pass from the use of glucose to the use of free fatty acids. This type of nutritional approach, therefore, has a solid physiological and biochemical basis and is able not only to induce an effective weight loss, but is also an improvement of the different parameters related to cardiovascular risk [3,6]. 

Among the different ketogenic diets, the one with very low calorie (very low caloric ketogenic diet, VLCKD), provides an intake of less than 50 g/day carbohydrates. From the moment when the ketogenic diet reduces the carbohydrate intake, on the one hand, there is a reduction in the production of insulin that normally has a liposynthesis activity, and on the other, the glucagon levels increase, which has an activity opposite to insulin, neoglucogenesis, and lipolysis; therefore, the catabolism of the reserve triglycerides is activated, which, by means of the intra-adipocitary lipases, undergo a hydrolysis and are transformed into glycerol and fatty acids [7]. 

It has also been observed that such ketogenic diets offer additional metabolic advantages: in addition to preserving muscle mass, they reduce appetite as ketone bodies have an anorexic effect; increase basal metabolism thanks to the preservation of protein and fat loss; and induce metabolic activation of thermogenesis. These diets also promote a non-atherogenic lipid profile, and reduce liver volume, blood pressure, and insulin resistance, with an improvement in blood glucose and insulin levels [8,9,10]. Ketones cross the blood–brain barrier and represent a more efficient glucose fuel, confirming the beneficial effects on cognitive functions obtained in patients with Alzheimer’s disease [11,12]. Although the effects of ketosis on the cognitive functions of healthy subjects are still poorly investigated, significant biochemical evidence suggests, however, that ketones may have a positive impact on brain functions. 

This improvement in brain function can also be achieved by exploiting the antioxidant properties of ketones [13,14]. The SNC participates in the control of energy homeostasis, acting on the regulation of its different components (contribution, expenditure, and caloric storage). The sympathetic nervous system (SNS) is a key factor in the regulation of the energy balance, acting both centrally on the regulation of the sense of hunger and satiety, both through the regulation of energy expenditure in peripheral tissues. Recent experimental evidence indicates that this system may play a role in the pathogenesis of obesity. 

In addition, the influences of a ketogenic diet on brain function are reported by the data literature. Salivary amylase is modulated by central nervous system and its production is ameliorated by VLCKD through modulation of the central nervous system [13]. As is well known, the autonomic nervous system also has an effect on the frequency of heartbeats, and the physiological variation between these heartbeats is used as an indirect measure of autonomic nervous system activity, which is known as heart rate variability (HRV) and is a parameter used to evaluate sympathetic and parasympathetic modulation of the heart.

Heart Rate Variability (HRV) is the natural variability of heart rate in response to several factors. HRV is related to the interaction between the SNS and the Parasympathetic. Our body, always, is in a concordant situation by the balance or by the predominance of one of these two nervous systems. HRV is an indirect and non-invasive method, based on the modification of the R-R intervals extrapolated from the electrocardiogram analysis [15,16]. In light of this evidence, this study intends first of all to verify how a nutritional intervention, based on caloric restriction and on the induction of a paraphysiological ketosis (VLCK diet), can obtain a reduction in BMI, mainly caused by the reduction in fat mass, comparing the results with those obtained by taking a conventional low-calorie diet (LCD diet).

In particular, the aim of this study is to research the possible functional changes, induced in the short term by the two dietary regimens, on the functions of the nervous system, with particular attention to the autonomous control of heart rate variability (HRV). The effects of the ketogenic diet on the cardiac response can also provide useful information about the role that the ANS plays on the control of visceral lipids and how an energy metabolic capacity can alter this control function, as well as other functions which see the SNA largely involved.

## 2. Materials and Methods

### 2.1. Subjects Recruitment 

In the present research work, a total of 26 subjects were recruited at the Department of Experimental Clinical Medicine of the University of Foggia, aged between 23 and 56, of both sexes, and obese (BMI > 30 kg/m^2^). The study was approved by the local Ethics Committee 22 May 2018, n°440/DS, and conducted according to the ethical principles of the Declaration of Helsinki. Written informed consent was obtained from all participants. These subjects underwent diet therapy in order to reduce body weight; they were also randomly divided into two groups: the first group (VLCKD, Very Low Caloric Ketogenic Diet) included 13 subjects (mean BMI at T0 34.48 ± 3.85 kg/m^2^), who were administered a VLCK diet, following the instructions provided by the “Lignaform” weight loss program (Laboratoire Therascience, Monaco-Ville, Monaco); the second group (LCD, Low Caloric Diet), on the other hand, included 13 subjects (average BMI at T0 33.4 ± 2.08 kg/m^2^), subjected to a conventional low-calorie diet; both diets have the same calorie intake (900 kcal). 

The fundamental requirement was that the subjects to be enrolled were in a good condition of health, i.e., in a condition of pre-diabetes and/or first-degree hypertension, according to the guidelines of the ESH/ESC, 2007 (140–159/90–99 mmHg). In addition, 14 subjects were excluded in based on exclusion criteria such as: renal insufficiency, severe hepatic insufficiency, type 1 or 2 diabetes mellitus treated with insulin, atrio-ventricular block, heart failure, cardiovascular and cerebrovascular diseases, unbalanced hypokalaemia, hypo-hyperthyroidism, chronic treatment with corticosteroid drugs, severe mental disorders, neoplasms, pregnancy, and lactation.

### 2.2. The Two Diet Programs

A total of 13 subjects were recruited for LCD diet programs. Participants in this group were assigned a balanced diet based on the following subdivision of macronutrients: 45–55% of carbohydrates, 15–25% of proteins, and 25–35% of fat; in addition, a dose of 20–40 g/day of fiber, in the form of fruits and vegetables, was recommended.

The subjects recruited into the VLCKD group are 13 and followed a ketogenic diet based on the introduction of high biological value protein preparations and natural foods, according to the commercial weight loss program “Lignaform Method” (Therascience). Each protein preparation contains an average of 18 g of protein. The method involves 3 phases: the ketogenic phase, the glucose reintroduction phase, and the food balance phase. The first phase involves the assumption of a low-calorie diet (600–800 kcal/day), low in carbohydrates (<50 g/day of vegetables) and lipids (30 g/day of olive oil); as regards, instead, the quantity of proteins with a high biological value, it is calculated on the value of the ideal body weight, multiplied by a coefficient of 1.2 for the woman and 1.5 for the men, in order to avoid the loss of lean body mass. The ketogenic phase was, in turn, divided into a first phase, defined as the “active phase”, in which the subjects took 4/5 times a day the protein preparations with a high biological value and vegetables with a low glycemic index, and a second phase, defined as the “selective phase”, in which one of the daily protein portions is replaced with a protein meal consisting of natural foods (e.g., meat, fish, eggs). During the ketogenic phase, the subjects were provided with supplements based on vitamins and minerals (K, Na, Mg, Ca) and omega-3 fatty acids (1 g/day), in accordance with international recommendations. The ketogenic phase was maintained up to the achievement of 80% of the predetermined weight loss goal (10% of the starting weight); moreover, the duration of this first phase varied between 45 and 60 days, depending on the pre-established weight loss objectives and the conditions of each participant.

In the second phase, the subjects begin to follow a low-calorie diet (LCD), through a progressive reintroduction of different groups of foods containing carbohydrates (in order: fruit, milk and derivatives, bread, pasta and legumes). This glucose reintroduction phase ends when the weight loss goal is reached, no later than 6 months from the start of the study.

Subsequently, in the third and last phase, all the subjects participating in the study were suggested to observe a dietary balance program, aimed at maintaining the lost weight in the long term.

### 2.3. Stages of the Study

For both groups, the trial, lasting a total of six months, was divided into 3 times:Time T0: initial phase in which the subjects are recruited and assigned to one or the other group. The subjects undergo a general medical examination and dietary evaluation by completing the food diary; instrumental evaluation of anthropometric parameters: BMI by weight measurement (b ilancia SECA, Tecnomed s.a.s., Verona, Italy, maximum capacity 150 Kg and approximation not exceeding 100 g) and height (SECA 711 stadiometer, Tecnomed s.a.s., Verona, Italy), body circumferences, skin folds; blood chemistry profile; analysis of the oxidation state by evaluating the plasma dosage of free radicals and endogenous antioxidant agents; electrocardiographic recording (ECG) and measurement of heart rate variability (HRV).T1 time: period necessary to achieve weight loss by administering the appropriate diet. During this phase, subjects must reach 80% of the expected weight loss goal and, on a weekly basis, instrumental body composition checks are carried out by means of BIA, instrumental evaluation of anthropometric parameters and, only for the VLCKD group, verification of the permanence of the subjects in the condition of dietary ketosis is carried out by monitoring the plasma levels of β-hydroxybutyrate. At the end of this phase, the subjects of both groups are subjected to the repetition of the whole series of blood chemistry and instrumental tests proposed in phase 1.Time T2: subjects belonging to both groups continue to follow the hypocaloric dietary regimen assigned to them until the predetermined weight loss is achieved. Even at the end of this phase, the subjects are subjected to the repetition of the battery of blood chemistry and instrumental tests carried out in the first two phases. Furthermore, in this last phase, all subjects are prescribed an aerobic exercise program at 60–70% of maximum heart rate, lasting 60–90 min and twice a week.

### 2.4. Heart Rate Variability (HRV)

Heart rate variability (HRV) was assessed at times T0, T1, and T2 in all participating subjects using an electrocardiogram (ECG). Before recording the electrocardiographic trace, the subjects were asked to stay in a quiet room for about 10 min. Two electrodes were then placed on the subject’s chest. ECG activity was recorded for at least 5 min at a sampling rate of 500 Hz, using the BrainVision acquisition system (Brain Products GmbH, Gilching (Munich), Germany). People were asked to breathe normally while in a sitting position.

The HRV analysis was then performed using the Kubios HRV Version 2.2 software (Biosignal Analysis and Medical Imaging Group, Department of Applied Physics University of Eastern Finland, Kuopio, Finland). Time domain and frequency analyses were performed in accordance with the guidelines of the Task Force of the European Society of Cardiology and the North American Society of Electrophysiology. The time domain indices include: the mean interval RR (MeanRR); the standard deviation of the RR intervals (SDNN), which mainly evaluates the sympathetic activity of the long-term component of the HRV; the mean square root of successive differences in the RR interval (RMSSD), evaluating the short-term component of HRV, corresponding to parasympathetic activity.

Frequency domain indices include Low Frequency (LF, 0.04–0.15 Hz), reflecting simultaneous sympathetic activation and parasympathetic modulation; High Frequency (HF, 0.15–0.40 Hz), which is a measure of vagal control; Total Power (TP), which reflects the overall autonomic activity; the LF/HF ratio, which is a measure of the sympathovagal balance.

### 2.5. Statistical Analysis

Data were analyzed using Microsoft software, version 6.01 for Windows (Microsoft, Redmond, WA, USA). Analysis of variance for repeated measures (ANOVA) with post hoc Tukey HSD (Honestly Significant Difference) was used to determine the differences between the dependent variables. A post hoc test (Bonferroni) for multiple comparisons was performed to identify significant differences between groups. All data were reported as mean ± SE. Statistical significance was considered for *p* ≤ 0.05. 

## 3. Results

Obese patients (BMI > 30) underwent two different diets: VLCKD and LCD. As shown in Figure 1, both diets resulted in a statistically significant reduction in BMI values, based on the three times analyzed (T0, T1, T2). In Table 1 are reported the principal characteristics of VLCKD and LCD obese subjects before and after diet. 

Analysis of variance showed significant effect both in the group following a VLCKD diet (F (2.25) = 0.00000671, *p* < 0.01) and in the group following the LCD diet (F (2.25) = 0.001, *p* < 0.01). In addition, we found a statistical difference between T0 and T1 in both VLCKD and LCD (*p* < 0.01) but not between T1 and T2.

Analyzing the heart rate (HR) in the two groups under examination (Figure 2), it emerges that, although in both groups the frequency trend appears almost similar (lowering of the HR at T1 compared T2), the variability is statistically significant only in the group subjected to a ketogenic diet, as suggested also by the *t*-test (F (2.25) = 0.008, *p* < 0.01). These data, in agreement with what is described in the literature, reflects a higher HR value in obese subjects. In fact, if we compare the HR parameter with that of the BMI, we have a direct proportionality between the two values.

By evaluating the mean-RR parameter (temporal distance between two heartbeats, expressed in ms), the trend in the two groups is similar, with an increase at T1 and a reduction at T2 (Figure 3). However, this variability is statistically significant only in the group subjected to a ketogenic diet (F (2.25) = 0.003, *p* < 0.01).

Analyzing the HF parameter (Figure 4), we find, as expected, a trend similar to that of the RR variability, with a significance in the ketogenic diet group (F (2.25) = 0.002, *p* < 0.01).

Instead, by evaluating the sympathetic activity through the analysis of the LF (Figure 5), its variability is closely correlated with the HR: in fact, the trend is similar with a reduction at T1 and an increase in values at T2. However, this variability is statistically significant only for the group subjected to VLCKD (F (2.25) = 0.023, *p* < 0.05).

## 4. Discussion

This study intends to evaluate the impact of weight loss induced by a low carbohydrate diet (VLCKD) and a conventional low-calorie diet (LCD) on the modulation of the cardiac autonomic nervous system and on the production of free radicals in obese subjects.

ANS makes an important contribution to the regulation of both the cardiovascular system and energy expenditure and is presumed to play a role in the pathophysiology of obesity and related complications [17,18,19]. Therefore, obesity and the autonomic nervous system are intrinsically related: a 10% increase in body weight is associated with a decline in parasympathetic tone, accompanied by an increase in average heart rate; conversely, weight reduction determines the decreased heart rate [20].

Furthermore, signals related to food intake from various origins (e.g., intestine, hepatoportal area, baroreceptors) are integrated at the level of the CNS and lead to an increase in SNS at the peripheral level. In fact, the reduction in food intake and the loss of body weight leads to a reduction in sympathetic activity. On the other hand, the pathophysiological mechanisms of obesity involve alterations of the sympathetic nervous system in accordance with the “Mona Lisa Hypothesis” (Most Obesities kNown Are Low in Sympathetic Activity). Furthermore, parasympathetic influences on energy expenditure demonstrate that an increase in parasympathetic modulatory activity can induce a paradoxical improvement in energy consumption [21,22].

Analyzing the results, in both groups we found a reduction in heart rate, as an indicator of sympathetic activity, following the decrease in weight (T1) compared with the baseline (T0). Consequently, always concurrently with the reduction in BMI, we observed an increase in the mean-RR parameter and an equally increase in HF (High Frequency), parasympathetic activity, synonymous with an increase in cardiac vagal modulation. From these results, we can deduce that weight loss is associated with an improvement in autonomic modulation through the enhancement of parasympathetic modulation, which clinically results in a decrease in heart rate. This is important because a higher heart rate is associated with increased mortality [23,24,25,26]

Furthermore, our study showed, in accordance with the studies cited above, that sympathetic activity (Low Frequency) is reduced in conjunction with caloric restriction and weight loss.

However, all observed parameters show a statistically significant variation only in the ketogenic diet group. Therefore, this study supports the notion that the sympathovagal balance can be modulated by a specific diet.

Furthermore, it is interesting to note how the values obtained in the VLCKD group at time T2 (time at which the subjects reintroduced all foods containing carbohydrates into their diet) return to be comparable to those obtained at time T0. From these data, we can deduce that it is probably the condition of ketosis that determines the significant modulation of the variables [27,28,29,30].

## 5. Conclusions

In conclusion, our data indicate that the functional modulation of the autonomic nervous system in the obese is improved after weight reduction, particularly in subjects undergoing dietary ketosis. Considering these results, we can affirm that the peculiarity of the study was to underline the difference in autonomic activity induced by the two different dietary regimes, and it represents a strong point of our study. Considering these results, it could be deduced that the ketogenic diet may have effects on the increase in antioxidant capacity [31] and heart rate variability (HRV).

However, the number of participants represents a limitation of the study; for this reason, further studies should be directed to reveal new aspects of the control exerted by the autonomic nervous system and the possible variation of oxidative stress in a greater number of subjects, so that they can be used for the prevention and treatment of obesity.

## Figures and Tables

**Figure 1 ijerph-19-02253-f001:**
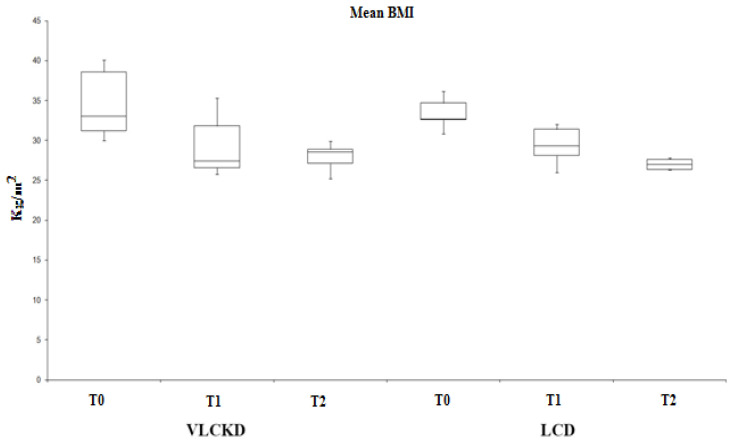
VLCKD and LCD diets induce a reduction in BMI values with statistical significance (*p* < 0.01).

**Figure 2 ijerph-19-02253-f002:**
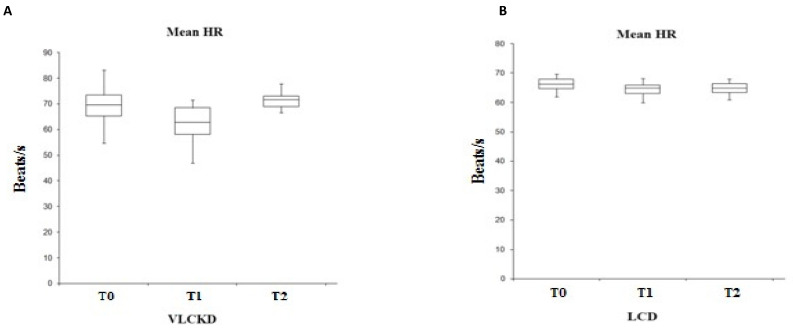
Variability of the mean HR in VLCKD subjects (**A**) and in LCD subjects (**B**).

**Figure 3 ijerph-19-02253-f003:**
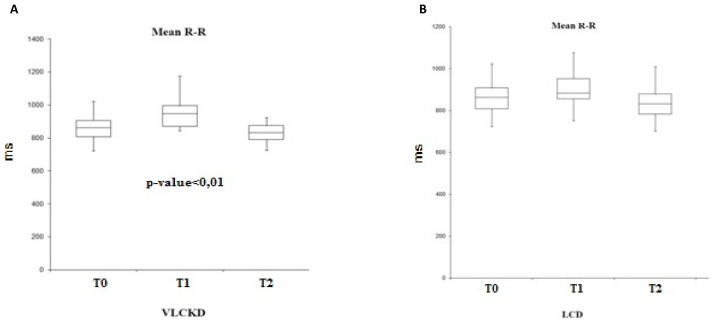
Variability of the mean R-R in VLCKD subjects (**A**) and in LCD subjects (**B**).

**Figure 4 ijerph-19-02253-f004:**
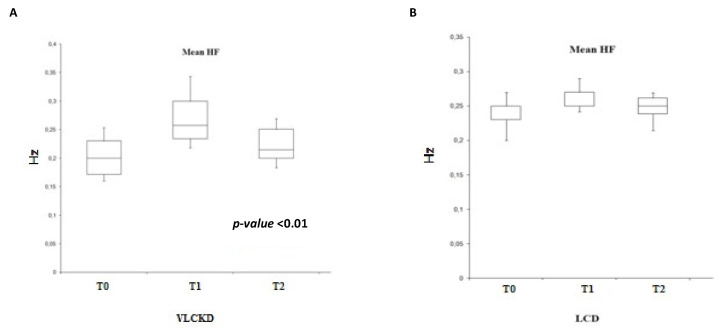
Variability of the mean HF in VLCKD subjects (**A**) and in LCD subjects (**B**).

**Figure 5 ijerph-19-02253-f005:**
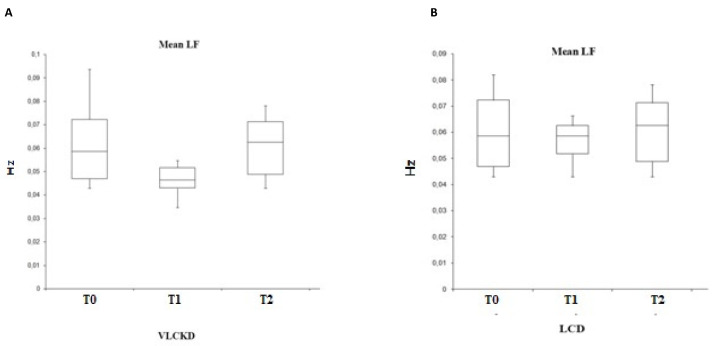
Variability of the mean LF in VLCKD subjects (**A**) and in LCD subjects (**B**).

**Table 1 ijerph-19-02253-t001:** Main Characteristics of participants.

	VLCKD OBESE SUBJECTS			LCD OBESE SUBJECTS			
	T0	T1	T2	T0	T1	T2	*p*-Value
Participants	7/6			7/6			
(M/F)							
Age	48 ± 3			47 ± 3			ns
Height (m)	1.67 ± 5			1.67 ± 3			ns
Weight (kg)	91.33 ± 11	78.73 ± 9	77.8 ± 5	86.33 ± 7	79.3 ± 8.7	76.28 ± 9	<0.001
BMI (kg/m^2^)	34.48 ± 5	27.7 ± 4	27.25 ± 6	33.4 ± 8	28.6 ± 4.5	26.85 ± 4	<0.001
Blood Ketones (MMOL/L)		1.9 ± 0.05	0.6 ± 0.2	0	0	0	<0.001

## Data Availability

Data is contained within the article. Authors can use this data for research purposes only by citing our research article.

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
