# Peer review of "Heart Rate Variability and Sympathetic Activity Is Modulated by Very Low-Calorie Ketogenic Diet"

_ijerph, 2022, doi:10.3390/ijerph19042253_

Round 1
Reviewer 1 Report
In this article, the authors compared the difference of heart rate variability between two groups of subjects with obese: one with low calorie ketogenic diet and the other with low caloric diet. The results are interesting. Here are a few questions and comments I have:
- 26 subjects were randomly divided into two groups. When sample size is not very large, it is not a good method to divide subjects into two groups randomly. A better way is stratified randomization. For example, there are both male and female in this study. We can randomly divide male into two groups and female into two groups to make sure each group has similar male/female proportion. There might be more male in one group if 26 subjects were randomly divided. Although BMI of each group is listed in the main text, it is better to include a table to show basic information within the two groups like BMI, sex, age, etc.
- One group of subjects took LCD and the other group took VLCKD. As far as I know, LCD is a diet between 800-1200 calories, while VLCKD is a diet less than 800 calories. The difference we saw in the paper might due to calorie difference instead of keto difference. I am not sure why the authors used LCD instead of VLCD.
- The authors used ANOVA to compare difference among 3 stages. ANOVA only tests whether there are any difference among groups. In this study, if we get a small p value from ANOVA, we only know there is significant difference between AT LEASE one pair of stages. It could be T0 and T1, T0 and T2, or T1 and T2. We don't know which one it is. We cannot conclude that " lowering of the HR at T1 and subsequent rise at T2" in Figure 2 from ANOVA. The authors should take other tests to support this conclusion. Similar issues for Figure 1 and some other figures. For example, in Figure 1, small p value of ANOVA from testing BMI difference between different stages in VLCKD group shows reduction of BMI between T1 and T0. There should be no reduction between T2 and T1.
- Bonferroni method was used for multiple test correction? Are all the p values here after correction or not?
- "Multivariate regression analysis was performed " (line 216), I did not see any results from the regression analysis.
- line 221, Fig. 10 should be Fig 1.
Author Response
In attached response point by point.
thank you

Reviewer 2 Report
1. 2.1. Subjects recruitment section.
How many people were excluded by the exclusion criteria listed in line 125, respectively? In addition, did any participants drop out during the program in the VLCKD group or LCD group? This information is important to assess selection bias.
2. 2.5. Statistical analysis.
Please change the software company name to GraphPad Prism's company name instead of Microsoft.
3. Results.
First of all, please show the baseline characteristics for the VLCKD group and the LCD group in Table. This table should include age, sex, height, weight, BMI, and any other available information.
4.
Page 5, line 221 "Fig. 10" appears to be an error for "Fig. 1".
5. Figure 1. and later.
The expressions of "Fig." and "Figure." is mixed. Please unify to either.
6. Figure 1. and later.
Since the graph shows the transition of values from T0 to T2, it would be easier to understand if it were a line graph.
7.
In Line 180, the authors state that they measured plasma ketone bodies of participants in the VLCKD group. What was the result of this? The value of the present study will be further enhanced if they can prove that ketosis existed among the participants of the VLCKD group. If there were no ketosis among the participants of the VLCKD group, it is better to weaken the expression of line 301-303.
8. Discussion.
Please state the strengths and limitations of the present study. Has there been a similar report before? If so, what was new or better than previous studies? Please state these points as strengths.
Since the number of participants in this study was relatively small, it would have been difficult to further subdivide the results (eg, by gender). Please list such points as limitations.
Author Response
In attached response point by point
thank you

Reviewer 3 Report
I recommend to accept the manuscript after minor revision.
There are only some points to correct:
- please provide the list of abbreviations
- please provide the number of ethical approval
- introduction and discussion section need improvement; please provide information on how your results will translate into clinical practice
- in discussion section please provide study strong points and study limitation section
- please correct typos
All abovementioned issues are crucial for the credibility of the results. The paper can be accepted only after addressing all the issues and another subsequent review.
I recommend to accept the manuscript after minor revision.
Author Response
in attached response point by point
thank you

Round 2
Reviewer 1 Report
The authors have addressed all my comments except one.
We can get small p values for both figure 1 and 2 in VLCKD group. The authors concluded in figure 2 that "lowering of the HR at T1 and subsequent rise at T2" because p value from ANOVA is significant. P value in figure 1 is also significant from ANOVA in VLCKD group. Can we also conclude that "lowering of the BMI at T1 and subsequent rise at T2 in VLCKD group"? I do not think so. BMI does not have a significant rise at T2 in VLCKD group. We cannot conclude "lowering of the HR at T1 and subsequent rise at T2" from significant p value of ANOVA. Some other tests should be applied here.
Author Response
In attached response to reviewer.
thank you
